# FDG-PET versus Amyloid-PET Imaging for Diagnosis and Response Evaluation in Alzheimer’s Disease: Benefits and Pitfalls

**DOI:** 10.3390/diagnostics13132254

**Published:** 2023-07-03

**Authors:** Poul F. Høilund-Carlsen, Mona-Elisabeth Revheim, Tommaso Costa, Kasper P. Kepp, Rudolph J. Castellani, George Perry, Abass Alavi, Jorge R. Barrio

**Affiliations:** 1Department of Nuclear Medicine, Odense University Hospital, 5000 Odense C, Denmark; 2Research Unit of Clinical Physiology and Nuclear Medicine, Department of Clinical Research, University of Southern Denmark, 5230 Odense M, Denmark; 3The Intervention Centre, Division of Technology and Innovation, Oslo University Hospital, 0372 Oslo, Norway; monar@ous-hf.no; 4Institute of Clinical Medicine, University of Oslo, 0313 Oslo, Norway; 5GDS, Department of Psychology, Koelliker Hospital, University of Turin, 10124 Turin, Italy; tommaso.costa@unito.it; 6FOCUS Lab, Department of Psychology, University of Turin, 10124 Turin, Italy; 7Section of Biophysical and Biomedicinal Chemistry, DTU Chemistry, Technical University of Denmark, 2800 Kongens Lyngby, Denmark; kpj@kemi.dtu.dk; 8Department of Pathology, Feinberg School of Medicine, Northwestern University, Chicago, IL 60611, USA; rudolph.castellani@nm.org; 9Department of Neuroscience, Developmental and Regenerative Biology and Genetics of Neurodegeneration, Departments of Psychiatry and Neuroscience, University of Texas at San Antonio, San Antonio, TX 78249, USA; perry2500@gmail.com; 10Department of Radiology, Perelman School of Medicine, University of Pennsylvania, Philadelphia, PA 19104, USA; abass.alavi@pennmedicine.upenn.edu; 11Department of Molecular and Medical Pharmacology, David Geffen UCLA School of Medicine, Los Angeles, CA 90095, USA; jbarrio@mednet.ucla.edu

**Keywords:** Alzheimer’s disease, Aβ, amyloid, amyloid-PET, FDG-PET, ARIA, MRI

## Abstract

In June 2021, the US Federal Drug and Food Administration (FDA) granted accelerated approval for the antibody aducanumab and, in January 2023, also for the antibody lecanemab, based on a perceived drug-induced removal of cerebral amyloid-beta as assessed by amyloid-PET and, in the case of lecanemab, also a presumption of limited clinical efficacy. Approval of the antibody donanemab is awaiting further data. However, published trial data indicate few, small and uncertain clinical benefits, below what is considered “clinically meaningful” and similar to the effect of conventional medication. Furthermore, a therapy-related decrease in the amyloid-PET signal may also reflect increased cell damage rather than simply “amyloid removal”. This interpretation is more consistent with increased rates of amyloid-related imaging abnormalities and brain volume loss in treated patients, relative to placebo. We also challenge the current diagnostic criteria for AD based on amyloid-PET imaging biomarkers and recommend that future anti-AD therapy trials apply: (1) diagnosis of AD based on the co-occurrence of cognitive decline and decreased cerebral metabolism assessed by FDA-approved FDG-PET, (2) therapy efficacy determined by favorable effect on cognitive ability, cerebral metabolism by FDG-PET, and brain volumes by MRI, and (3) neuropathologic examination of all deaths occurring in these trials.

## 1. Introduction

The amyloid cascade hypothesis postulates that Alzheimer’s disease (AD) is caused by cerebral deposits of the protein amyloid-beta (Aβ) [1,2]. Despite substantial criticism throughout its 30-year existence, the hypothesis is still alive and has served as rationale for several phase 2 and 3 randomized clinical trials (RCTs) aiming to document the efficacy of anti-Aβ AD interventions [3]. 

Among such interventions, monoclonal antibodies for passive immunotherapy have attracted particular interest since the US Food and Drug Administration (FDA) on June 7, 2021 granted accelerated approval for the drug aducanumab (also known as Aduhelm) produced by Biogen/Eisai. The approval was based exclusively on a presumed drug-induced removal of cerebral Aβ deposits—assessed by amyloid-positron emission tomography (amyloid-PET)—followed by the remark that this “is expected to lead to a reduction in the clinical decline of this devastating form of dementia” [4]. Further it was stated that “The use of a surrogate endpoint can considerably shorten the time required prior to receiving FDA approval” and also that “Drug companies are required to conduct post-approval studies to verify the anticipated clinical benefit. These studies are known as phase 4 confirmatory trials” [4].

On 6 January 2023, the antibody lecanemab (also known as Leqembi) by Eisai/Biogen was also granted accelerated FDA approval; again with referral to therapy-induced removal of cerebral Aβ, albeit, this time with the remark that “The results of a Phase 3 randomized, controlled clinical trial to confirm the drug’s clinical benefit have recently been reported and the agency anticipates receiving the data soon” [5]. FDA’s assessment of a third anti-AD antibody, donanemab, has been pending, awaiting additional trial data, which according to a recent press release from the producer, Eli Lilly, is expected in the near future. 

The phase 2 and 3 trials of these antibodies included patients with mild cognitive impairment (MCI) or mild AD only if “amyloid-positive” according to amyloid-PET using the tracer ^18^F-florbetapir; for lecanemab also ^18^F-florbetaben or ^18^F-flumetamol. All three antibodies indicate significant therapy-induced reductions in amyloid-PET signal during the trial periods. This observation has been assumed, without discussion of location, specificity, and molecular amyloid target, to reflect one-to-one reduction in the brain’s “amyloid burden”. However, as discussed below, we believe it is problematic to study only “amyloid-positive” patients, and more seriously, to assume that a reduced amyloid-PET tracer signal is 100% due to removal of amyloid, as implied by the surrogate approvals and interpretation of the data. In contrast, we argue that amyloid-PET is not 100% specific to a single molecular target and that decreased uptake of amyloid-PET tracer is at least partly due to increased therapy-related cell damage.

## 2. Clinical Efficacy

For aducanumab, there have been no press reports of significant clinical benefits, although a large phase 3 RCT, ENGAGE, involving over 1600 patients, showed significantly less impairment on the cognitive Clinical Dementia Rating-Sum of Boxes (CDR-SB) scale (range 0–18, with higher scores indicating greater impairment) in patients with MCI or mild AD dementia. However, the finding was negated by an almost identical trial, EMERGE, comprising as many of the same kind of patients, which could not reproduce any of the significant results observed in ENGAGE [6]. This lack of reproducibility in large randomized controlled trials is by itself important and carries relevance to large single trials of lecanemab and donanemab.

The clinical benefits of lecanemab and donanemab, claimed in press releases to indicate a “27% or 35% slowing of cognitive decline”, respectively [7,8], actually reflect modest absolute changes on the corresponding scales: With lecanemab, the 27% reduction reflected an increase of 1.21 points on the CDR-SB scale compared to 1.66 points with placebo, i.e., a difference of 0.45 points equal to 27% of 1.66 [9]. However, the 0.45 points equal only 2.5% of the scale range and are less than half of the 1–2 point increase that is considered the “minimal clinically important difference” (MCID) on the CDR-SB scale in this category of patients [10]. Furthermore, the 27% hides major unexplained heterogeneity in efficacy varying by many hundred percent across subgroups. This biologically implausible heterogeneity could relate to dropout tendency correlating with outcome via socioeconomic and health covariates. Unfortunately, authors tend to not provide dropout statistics and health covariates on subgroup levels. The substantial dropout of participants and the relatively small fraction of eligible subjects who are ultimately studied may cause considerable selection bias that tends to belie the claimed average efficacy.

The recent press release on donanemab mentions a 35% slowing of decline (compared to placebo) on the constructed cognitive/functional Integrated Alzheimer’s Disease Rating (iADR) scale (range 0–144, with lower scores indicating greater impairment) and a 36% slowing on the CDR-SB scale. These “slowings” compared with placebo correspond to what can be estimated to a less worsening of about 3.2 points on the iADR scale and about 0.5–0.6 points on the CDR-SB scale [11], i.e., changes of approximately ⅓ and ½, respectively, of the MCID on these two scales [10,12].

We are not convinced that these few significant cases of small and clinically insignificant delays in cognitive and functional decline are real, as the heterogeneity and dropout reasons indicate sources of bias, e.g., due to withdrawal of some patients with amyloid related imaging abnormalities (ARIAs, see later) with and without symptoms, or attrition bias from depletion of negative health covariates in the treatment arm correlating with dropout tendency (e.g., efficacy was by far largest for male Americans, and much smaller for female Europeans). ARIA could also lead to functional unblinding of a large number of participants, introducing further bias. Thus, the trials lack information of crucial importance on combined subgroup statistics for dropout, and exactly how many patients were censored on which grounds, whether unblinding influenced neurocognitive outcome assessment, and whether these features influence the efficacy estimates.

Given these uncertainties, it is understandable that FDA in its accelerated approvals put emphasis on a surrogate biomarker, specifically, the assumed removal of amyloid deposits as assessed by amyloid-PET. Unfortunately, however, in our opinion this choice is also problematic, as we explain below.

## 3. PET Imaging in General

The many trial reports leave an impression that the researchers perceive PET much in line with CT and MRI, although this is a misunderstanding. It thus seems appropriate to give a brief description of the PET methodology, its advantages, and disadvantages.

CT and MRI are high spatial resolution modalities showing disease-related tissue changes, but only when the disease has developed for some time. In contrast, PET has poorer spatial resolution, but much higher sensitivity (by a factor of 1000 or more) [13]. Being a molecular modality, PET can detect disease earlier (weeks, months, years) than CT and MRI, i.e., when the disease is hypothesized to be more sensitive to therapy. Furthermore, PET is inherently quantitative, meaning that it can in principle measure not only focal disease activity but also the overall disease burden in an organ or in the body and monitor change in this as a measure of therapy efficacy [14,15]. Therefore, hybrid PET/CT and PET/MRI scanners are appropriately combined imaging modalities, since in a single examination séance, PET can detect and measure early-stage disease, while CT or MRI can show more precisely where the disease is located. In general, the combination increases the chance of early and accurate diagnosis. This is beneficial in daily patient management as well as in RCTs testing new therapeutic options.

PET scans do not provide a snapshot, but a quantifiable “movie” with high time resolution of the physiological or pathophysiological process to which the body exposes the tracer in question [16]. This means that proper utilization and interpretation of the scan requires—in addition to disease knowledge—a thorough understanding of the many facets involved. These include (1) the properties of the tracer in use (administration, targeting, affinity, kinetics, metabolism, etc.); (2) experience with acquisition techniques and image data processing and quantification, all of which are less complicated with CT and MRI, for which the manufacturing companies typically provide an automated set of results that the reviewer has limited ability to modify; and (3) interpretation of the validity of the acquired amyloid-PET signals, whether they are reliable or may be due to false targeting or circumstances unrelated to the disease at hand, including non-specific incidental findings and varying blood background activity, the magnitude of which has a significant impact on the signal-to-noise ratio, not least when calculating SUVr. 

All this, together with expensive equipment and costly synthesis development and testing of new PET tracers not to mention the need of an on-site cyclotron and radiochemical laboratories explains not only the sparse use of PET compared with CT and MRI, but also the often inadequate understanding of the far greater possibilities of the PET method—simply because so relatively few healthcare professionals have had the method in their hands and gained experience with its possibilities and qualities. Hopefully, this may change with the arrival of total-body PET scanners with much higher (in practice 10 times) sensitivity and increased use of artificial intelligence to accelerate image processing and make results more reliable [17]. However, a remaining concern is that PET tracers may have weaknesses in terms of specificity and mode of action. These issues must be fully clarified before the tracers are used for diagnostics and response evaluation in large clinical RCTs. However, as far as amyloid tracers are concerned, they are insufficiently elucidated, and it is still unclear exactly which components inside and outside the cells they target in vivo and whether they target other beta-sheet structures [18,19]. 

## 4. The Diagnosis of AD or Alzheimer’s Syndrome

### 4.1. Before the Advent of Amyloid-PET Imaging

It is fair to say that no one knows for sure what AD is, and, therefore, no one can tell how to diagnose it with any certainty. Nor does anyone know for sure whether AD is one or more diseases or perhaps rather a syndrome, i.e., a symptom complex caused by one or more factors, so that the abbreviation AS may be more appropriate than AD. Others have attempted the same distinction in talking about “Alzheimer’s clinical syndrome” [20]; however, they have been reprimanded by the rule makers with the remark: “This is a radical statement that simply does not represent the longstanding definition of Alzheimer’s in the field” [21].

Alois Alzheimer did not describe any ‘disease’ under his name. He started in 1901 making clinical observations of the 50-year-old Frau Auguste Deter, who was progressively confused and forgetful until her death in 1906, when at autopsy he found “senile plaques” and neurofibrillary tangles in the gray matter of her shrunken brain, which, however, he did not relate to cerebral amyloid deposits in the paper he published about his findings [22], even though he had discussed hyaline-like findings in the vicinity of cerebral vessels at autopsy 10 years earlier of two younger adults suffering from paralysis and seizures, respectively [23]. Alzheimer’s mentor, Emil Kraepelin, a preeminent psychiatrist at the time, apparently was the first to introduce the term “Morbus Alzheimer”, perhaps as an acknowledgment of Alzheimer’s work, which was otherwise met with little interest and no questions from colleagues when he presented his now famous observations of Auguste Deter at a scientific conference [24]. Unfortunately, however, equating the term Morbus Alzheimer to AD may be an unfortunate over-simplification.

Amyloid deposits in the body and in the brain have been described since the 17th century [25,26]. However, according to Knopman et al., the designation “AD” did not appear as a stand-alone diagnosis, until in the hospital version of the 1975 International Classification of Diseases adopted by the Mayo Clinic, where the term “AD” was a codable diagnosis. Since then, also according to Knopman et al., the definition has changed comprising three successive phases with AD being defined as (1) a clinicopathologic entity, (2) a postmortem pathobiological entity, and, for the time being, (3) an antemortem pathobiological entity [27]. 

The first stage started in 1984 with the addition of the prefix “probable” to AD to indicate that the assignment of etiology was provisional, while the term “definite” was reserved for cases with autopsy confirmation. The shift to the second diagnostic model was partly justified by the insufficient specificity of the first, which became an almost purely clinically based approach due to limited access to neuropathology services. This was followed in 1997 by a shift to a neuropathological model, based on the combination of significant presence of neuritic plaques and neurofibrillary tangle pathology in post mortem brains. However, this model meant that the diagnosis was now applied to a substantial proportion of individuals who were cognitively normal (reviewed in [27]). 

### 4.2. After the Advent of Amyloid-PET Imaging

The advent of the first amyloid-PET tracer, ^11^C-Pittsburgh Compound-B (^11^C-PiB) in 2004, which was expected to “provide quantitative information on amyloid deposits in living subjects” [28], heralded the next change in that *antemortem* diagnosis with an image biomarker was now considered a viable possibility. However, as time has gone by, this vision has not been fulfilled. The trend was substantiated by the appearance of the ^18^F-labeled amyloid tracers with claimed high diagnostic accuracy [29,30,31,32,33,34,35], followed by reports of significant management changes due amyloid-PET imaging [36,37,38,39,40], even though nobody is able to determine what is right or wrong. It is only recently that the long-held belief of a high negative predictive value of amyloid imaging has been disproved [41,42]. On the whole, with regard to claimed diagnostic accuracy, all AD “diagnostic measures” suffer from some amount of circular reasoning [43], as there is no universal clear-cut definition of AD/AS, and thus no way either to make a definitive diagnosis due to lack of an independent and reliable reference to compare with. 

Nonetheless, amyloid-PET found its way into the diagnostic criteria, when the US National Institute on Aging and Alzheimer’s Association (NIA-AA) in 2011 for research purposes endorsed a diagnosis of preclinical AD with the presence of positive AD biomarkers (CSF or amyloid imaging) and the absence of cognitive impairment [44]. These and the 2011 criteria for the “diagnosis of dementia due to Alzheimer’s disease” [45] were systemized 7 years later based on a preceding “unbiased” work-up [46] by a large number of the authors, who created the 2018 NIA-AA Research Framework including the A/T/(N) classification. This scheme places crucial emphasis on “A”, biomarkers of Aβ plaques, i.e., “cortical amyloid ligand binding or low CSF Aβ_42_, and “T”, which is increased CSF phosphorylated tau (P-tau) and cortical tau PET ligand binding. In contrast, “(N)” are biomarkers of neurodegeneration or neuronal damage: CSF tau, FDG-PET hypometabolism, and atrophy on MRI, but have less significance, such that A+ is the crucial prerequisite for the diagnosis to fall within the “Alzheimer’s continuum”, whereas T+ or (N+) without concomitant A+ by definition means other non-AD pathology (Table 1) [47]. 

Some of these authors applied the 2018 NIA-AA Research Framework to a sample of participants in the Mayo Clinical Study of Aging, a population-based cohort study of cognitive aging in Olmsted County, Minnesota, including initially 5213 individuals aged 60–89 years, of which 1524 underwent amyloid-PET and 576 underwent both amyloid and tau PET. The authors found much higher prevalence of what they, according to the 2018 A/T/(N) scheme call ‘biological AD’ (i.e., A+/T+) and clinically defined ‘probable AD’: for women 10% vs. 1% at age 70 and 33% vs. 10% at age 85, and for men 9% vs. 1% and 31% vs. 9%, respectively, at the same two ages [48]. The only gender difference was a greater prevalence of MCI and dementia, assessed clinically, in men than women. This “biological vs. clinical” discrepancy made the authors conclude, verbatim: “These findings illustrate the magnitude of the consequences on public health that potentially exist by intervening with disease-specific treatments to prevent symptom onset”. 

Exactly how this is to be understood is difficult to grasp. One question is the real prevalence of AD, and how this changes depending on using the A/T/(N) scheme or not. With a more strict definition, the estimated number of 6.7 million Americans living with AD today [49] could be half that or even smaller. We therefore support the Centers for Medicare & Medicaid Services’ (CMS)’s decision to offer reimbursement for only one amyloid-PET scan per patient and advise it be maintained—despite the well-intentioned wishes of the Alzheimer’s Association and the African American community for increased access to these scans [49,50,51]—unless and until it is determined that passive immunotherapy is more beneficial than harmful. 

## 5. The Role of PET Imaging

### 5.1. Amyloid-PET Imaging

Importantly, the substance or pathobiological process that amyloid-PET tracers, with their different chemical profiles and targeting characteristics [18,19], actually engage in vivo is not clear. Thus, amyloid-PET signals are not an accurate reflection of amyloid in the gray matter, where the disease is located, as they target also inflammatory processes [52,53] and white matter hyperintensities, where there is no amyloid, as well as myelin and myelin damage [54,55,56,57,58,59]. 

The spill-over from the large white matter signal makes correct quantification of uptake in the narrow gray matter very difficult, if not impossible (Figure 1). In addition, all these trials use the standardized uptake value ratio (SUVr) as a measure for amyloid burden, where the ‘r’ indicates normalization with cerebellar activity, but pathology also exists in the cerebellum of these patients [60,61,62] meaning that the effect of these antibodies on amyloid removal cannot be judged from SUVr reduction [63].

Given the uncertainty in interpreting the declining amyloid curves in these trials, purportedly and perhaps misleadingly showing maximal reduction in amyloid burden of 71% with aducanumab [6], 55% with lecanemab [9], and 85% with donanemab [11], FDA approvals based on this surrogate endpoint should be revisited. The few PET images that have so far been published on this [64,65] show a large white matter signal that decreases during treatment, which we believe more likely is due to an almost universal, inflammatory cerebral reaction. This is consistent with the fact that these patients have a greatly increased frequency of the edema-type ARIAs (ARIA-E), which occur early and tend to subside during the course of treatment in most patients, as assessed by MRI. 

One might wonder if the curves instead show a natural removal of cerebral debris, including amyloid, caused by therapy-related tissue damage. If so, it could explain why immunotherapy has so little effect on the patients’ cognitive or functional ability. It is also more in line with the accelerated loss of brain volume that has been described as a result of anti-β-amyloid drugs [66] as also reported in some trials with the three antibodies discussed here [6,66] (Table 2). 

The assumption of therapy-related cell damage is further supported by a recent report of a patient with no vascular co-morbidities who developed stroke-like symptoms after three lecanemab infusions, and died of acute multifocal intracerebral hemorrhage following intervention with tissue plasminogen activator. Post mortem examination showed therapy-induced destruction of blood vessels (necrotizing vasculitis) involved by cerebral amyloid angiopathy [67]. 

### 5.2. FDG-PET Imaging

There is an excellent alternative to amyloid-PET and that is FDG-PET imaging, which is not widely recognized or used in this context, although the PET method was applied in this context long before the emergence of amyloid tracers, because what is more natural than investigating whether patients with suspected dementia actually have impaired cerebral function? The FDA early understood the value of the method as a diagnostic tool in Alzheimer’s disease, and CMS reimbursement was subsequently approved. The methodology has so far only been applied in two trials on anti-Aβ immunotherapy, i.e., a study of the antibodies gantenerumab vs. solanezumab and a trial of the antibody crenezumab; however, none of these trials indicated significant changes in the outcome measures including regional cerebral FDG uptake [68,69].

FDG-PET is the only well-known, tested and widely recognized method to assess global and regional cerebral metabolism of a measure of neuronal function [70], and there are recommendations on how the examination should be carried out [71]. Therefore, it should replace amyloid-PET imaging in dementia patients as part of the diagnostic work-up and in particular for long term monitoring of anti-AD treatments. *Only if a treatment can stop or reverse the decrease in FDG accumulation in the brain regions most affected in AD, i.e., the temporo-parietal cortices, there is reason to believe that the treatment has a clinically important effect* [72]. Moreover, it would be helpful to follow the patients with repeat MRI scans, not only to elucidate the occurrence of ARIAs but to measure changes in cerebral volumes to allow more thorough evaluation of how the therapy affects the brain. Based on the above considerations with regard to dementia diagnostics and therapy monitoring, we suggest a return to basics in the community and in future anti-Alzheimer’s trials as enumerated in four recommendations:

## 6. Recommendations

The diagnosis of AD/AS should be based upon co-occurrence of the following findings: (a) impaired cognitive function on the MMSE and/or CDR-SB scale, assessed by a trained neuropsychologist, (b) impaired temporo-parietal glucose metabolism assessed by FDG-PET according to standardized imaging and analysis procedures, (c) absence of other well-defined disorders, including tumor, metastases, trauma, and stroke, (d) absence of clinical disease phenotypes closely associated with frontotemporal lobar degeneration and young-onset dementia, excluded by standardized criteria, but not necessarily absence of vascular dementia.Positive therapy efficacy equals: (a) favorable change (exceeding an a priori predefined minimum limit) in cognitive ability as measured on a recognized cognitive scale, (b) increased global or specified regional cerebral metabolism assessed by repeat FDG-PET brain imaging, and (c) less decrease in global brain and hippocampal volumes and less increase in ventricular volume assessed by volume MRI.Registration and neuropathologic examination of all deaths occurring during and two years after termination of clinical trials should be carried out.The limited CMS reimbursement for amyloid-PET-scans in dementia patients should not be changed until results of phase 4 confirmatory immunotherapy trials are available.

## 7. Conclusions

We recommend a new critical look at methods and criteria currently used to diagnose dementia in living patients and to assess the efficacy of anti-dementia therapies. We suggest reinstatement of clinical evaluation of cognitive ability as the primary procedure assisted by FDG-PET, supplemented by amyloid-PET and volume MRI and independent neuropathologic examination of all trial death cases. We also recommend more detailed statistical analysis of subgroup effects, including study subjects who withdraw from clinical trials, and more rigorous risk of bias assessment than currently carried out in relevant trials. 

## Figures and Tables

**Figure 1 diagnostics-13-02254-f001:**
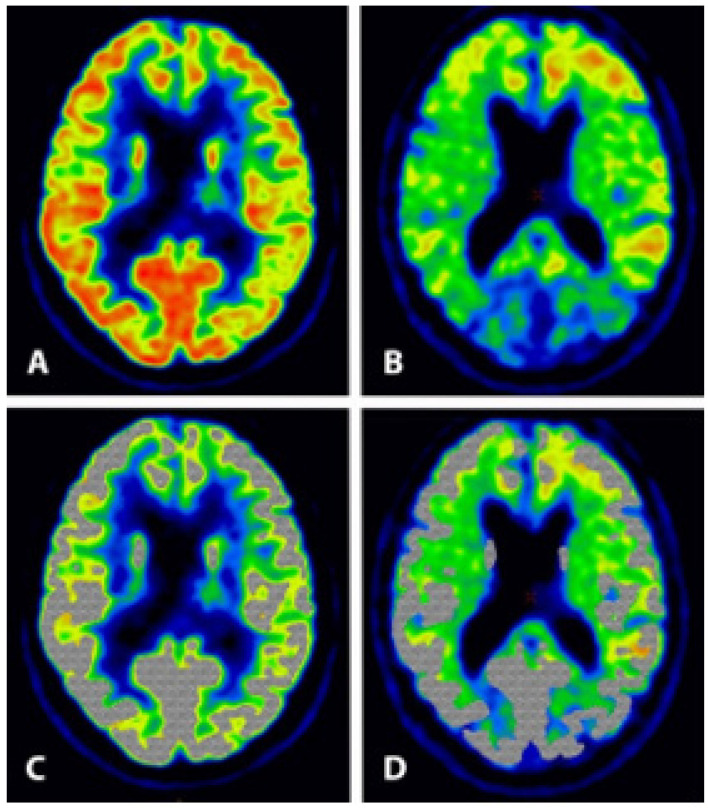
Patient in his early 70s with increasing memory impairment and loss of overview during some months. Clinical assessment including neuropsychological testing and FDG-PET suggested mild dementia—most likely AD type. Subsequent PiB-PET was positive. FDG uptake reflecting gray matter glucose metabolism is mainly situated in a narrow rim near the skull (**A**), whereas ^11^C-PiB uptake extends more centrally (**B**), indicating non-specific uptake in the white matter as illustrated when the shaded grey area, representing the FDG uptake (**C**), is superimposed on the ^11^C-PiB image (**D**). Grey:white matter uptake ratio was 4–5:1 for FDG and 1:1 for PiB.

**Table 1 diagnostics-13-02254-t001:** A/T/(N) Classification Scheme.

Biomarker Group	Biomarkers
A (Aβ plaques)	Cortical amyloid-PET binding Low CSF Aβ_42_ Low Aβ_42_/Aβ_40_ ratio
T (Tau)	CSF phosphorylated tau (P-tau) Tau-PET
N (Neurodegeneration/-injury)	Anatomic MRI FDG-PET CSF total tau

Aβ, amyloid-beta; CSF, cerebrospinal fluid; FDG, fluorodeoxyglucose; MRI, magnetic resonance imaging, PET = positron emission tomography.

**Table 2 diagnostics-13-02254-t002:** Findings in aducanumab, donanemab and lecanemab trials.

Drug, Author, Year, Trial Name	Subjetcs	Change on Clinical Scale T vs. P	Amyloid Change	ARIA-E T vs. P	ARIA-H T vs. P	Volume Loss
Aducanumab Sevigny et al., 2016 [64] PRIME	Prodromal or mild AD Positive Aβ PET 3 T vs. 1 P group	CDR-SB (0–18): 0.7 vs. 1.8 MMSE (0–30): −0.6 vs. −2.8	−27%	41% vs. 0%	9% vs. 5%	NR
Aducanumab Budd Haeberlein et al., 2022 [6], EMERGE	MCI due to AD or mild AD dementia Amyloid positivity	CDR-SB (0–18): 1.3 vs. 1.7 MMSE (0–30): −2.7 vs. −3.3	−71%	35%/26% vs. 2%	20%/16% vs. 7%	Yes
Aducanumab Budd Haeberlein et al., 2022 [6], ENGAGE	MCI due to AD or mild AD dementia Amyloid positivity	CDR-SB (0–18): 1.6 vs. 1.6 MMSE (0–30): −3.6 vs. −3.6	−58%	36%/26% vs. 3%	19%/16% vs. 6%	Yes
Lecanemab Swanson et al., 2021 [66] BAN2401-G000–201	MCI or mild AD Amyloid	ADCOMS (0–1.97): −0.14 vs. −0.18 CDR-SB (0–18): 1.1 vs. 1.5	−31%	9.9% vs. 0.8%	6.8% vs. 5.3%	Yes
Lecanemab Van Dyck et al., 2023 [9] Clarity-AD	MCI or mild AD Amyloid	CDR-SB (0–18): 1.2 vs. 1.7 ADCOMS (0–1.97): −0.16 vs. −0.22	−55%	12.6% vs. 1.7%	17.3% vs. 9.0%	NR
Donanemab Mintun et al., 2021 [11] TRAILBLAZER-ALZ	Prodromal AD (MCI incl.) and mild AD Amyloid positivity	iADRS (0–144): −6.9 vs. −10.1 CDR-SB (0–18): 1.2 vs. 1.6	−85%	26.7% vs. 0.8%	8.4% vs. 3.2%	NR

Aβ, amyloid-beta; AD, Alzheimer’s disease; ADCOMS, Alzheimer’s Disease Composite Score (higher score = greater impairment); ARIA-E, amyloid-related imaging abnormality—edema type; ARIA-H, amyloid-related imaging abnormality—hemosiderin type; CDR-SB, Clinical Dementia Rating-Sum of Boxes (higher score = greater impairment); iADRS, Integrated Alzheimer’s Disease Rating (lower score = greater impairment); MCI, mild cognitive impairment; MMSE, Mini-Mental State Examination (lower score = greater impairment); NR, not reported; P, placebo; T, treatment; vs, versus.

## Data Availability

Not applicable.

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
