# Peer review of "FDG-PET versus Amyloid-PET Imaging for Diagnosis and Response Evaluation in Alzheimer’s Disease: Benefits and Pitfalls"

_diagnostics, 2023, doi:10.3390/diagnostics13132254_

Round 1
Reviewer 1 Report
In the present paper the authors exhaustively and critically reviewed the amyloid cascade hypothesis and the use of amyloid PET as surrogate endpoint for the accelerated approval of some antibody, which had been shown to reduce amyloid burden.
The topic is of high relevance, and the authors developed it, from different perspectives, from imaging methodology to clinical and cognitive outcomes.
The paper is very well written, and the position of the authors clearly described.
I have only some minor suggestions:
Page 2 line 71: typo “18F-florbetapi”, please correct
Page 5 line 221: Please control ref 41 and 42 in the test and in the reference section
Page 6 line 257: please define CMS
Author Response
Response to Reviewer 1
We thank Reviewer 1 for the kind words.
The typo on page 2, line 71 has been corrected and on page 6, line 257 CMS is now spelled out.
We understand if the reviewer is a little confused about the references 41 and 42 in the text and in the reference list. Because it is confusing - and a bit of a long story:
I and my colleagues wrote in the spring 2022 an Opinion Paper/Commentary, which was published in Clinical Nuclear Medicine:
Aducanumab-Related Amyloid-Related Imaging Abnormalities: Paean or Lament?
Høilund-Carlsen PF, Werner TJ, Alavi A, Revheim ME. Clin Nucl Med. 2022 Jul 1;47(7):625-626. doi: 10.1097/RLU.0000000000004250. Epub 2022 Apr 22.PMID: 35452007
In response to that, Wassef & Colletti wrote the following Opinion Paper/Commentary also in Clinical Nuclear Medicine:
Commentary: Aducanumab-Related ARIA: Paean or Lament?
Wassef HR, Colletti PM.Clin Nucl Med. 2022 Aug 1;47(8):707-709. doi: 10.1097/RLU.0000000000004252. Epub 2022 May 11.PMID: 35543641
In response to that, we wrote the following Letter to the Editor of Clinical Nuclear Medicine (i.e., Patrick Colletti):
Re: Aducanumab-Related ARIA: Paean or Lament?
Høilund-Carlsen PF, Alavi A, Revheim ME.Clin Nucl Med. 2023 Jun 1;48(6):505-506. doi: 10.1097/RLU.0000000000004509. Epub 2023 Feb 1.PMID: 36724162
To which Wassef & Colletti responded with the following Opinion Paper/Commentary in Clinical Nuclear Medicine:
Re: Aducanumab-Related ARIA: Paean or Lament?
Wassef HR, Colletti PM.Clin Nucl Med. 2023 Feb 1;48(2):168-169. doi: 10.1097/RLU.0000000000004490. Epub 2022 Nov 11.PMID: 36607365
It sounds like there were a lot of disagreements, but no, it was quite the opposite. There was agreement on most, including that the negative predictive value of amyloid PET is limited, which is what the last two publications listed above are all about.
Technically, there came a “problem” in that our Letter to the Editor (reference 42) was written and submitted first, but published last, meaning that Wassef & Colletti's response (reference 41) to our letter (reference 42) was for some reason published before our letter.
So, we were in doubt as to which of the two references we should list first in the text and in the reference list. We chose to list them in chronological order of publication, even though the reality was the opposite. The order can easily be reversed in both places if preferred.
Very best,
Poul Høilund-Carlsen
Reviewer 2 Report
This paper is clear and well-written
According to the Authors guidelines, the in not a "perspective" section. Therefore, I suggest to classify this paper as a narrative review.
I suggest to modify the title in order to clarify that this review regards ONLY FDG and amyloid imaging or, alternatively, to include a dedicated section concerning Tau imaging and further PET tracers used in AD.
In suggest to use the EANM guide for appropriate nomenclature of radiotracers.
If you aim is to review the possible role of PET imaging in therapy asssessment I suggest to re-name the title of dedicated subparagraphs ( the role of PET imaging in treatment monitoring, for example) and I suggest to include more details in these sections.
I also suggest to improve the section concerning FDG imaging ( for example you can cite the following paper doi: 10.1111/ene.13728.
or further recent guidelines for the diagnosis with FDG)
Author Response
We thank Reviewer 2 for nice comments and suggestions.
We understand the Reviewer's first objection; however, the manuscript is written as a perspective article at the suggestion of Diagnostics' Managing Editor, Dennis Zhu, after a prior inquest and this has been accepted by Prof. Dr. Oke Gerke, who is Guest Editor of the special issue of Diagnostics that the manuscript is aimed at. Therefore, we have chosen to maintain the format as a perspective article, which is apparently a common format in several MDPI journals.
Regarding the title, we understand the reviewer's point and have chosen to follow the first suggestion, rewording the title to: “FDG-PET versus Amyloid-PET Imaging for Diagnosis and Response Evaluation in Alzheimer’s Disease: Benefits and Pitfalls”.
With regard to using the EANM's radiotracer nomenclature, we see that our manuscript has already been edited on this point as well. Therefore, we do not feel justified in changing the radionuclide nomenclature used, but believe that it should be up to Diagnostics Editorial Office to decide which nomenclature to use.
Finally, we do understand Reviewer 2's last two comments about expanding the explanation of FDG-PET in brain imaging and mentioning relevant recommendations and guidelines, including the Delphi consensus paper to which he refers. However, in our opinion, this belongs more in a review that goes in-depth on how these examinations should properly be performed. However, that's not the point of our perspective article, and it would be going too far. Our main purpose has been to point out that something is wrong and that amyloid-PET is an uncertain method for assessing AD and that FDG-PET should be used instead. Therefore, we only added a single sentence about this, including the reference mentioned by the Reviewer 2.
Best regards,
Poul Høilund-Carlsen